# Exploratory Quality Control Study for *Polygonum multiflorum* Thunb. Using Dinuclear Anthraquinones with Potential Hepatotoxicity

**DOI:** 10.3390/molecules27196760

**Published:** 2022-10-10

**Authors:** Huiyu Gao, Jianbo Yang, Xueting Wang, Yunfei Song, Xianlong Cheng, Feng Wei, Ying Wang, Donglin Gu, Hua Sun, Shuangcheng Ma

**Affiliations:** 1Institute for Control of Chinese Traditional Medicine and Ethnic Medicine, National Institutes for Food and Drug Control, Beijing 102600, China; 2School of Chinese Materia Medica, Beijing University of Chinese Medicine, Beijing 102488, China; 3Chinese Academy of Medical Sciences & Peking Union Medical College, Beijing 100050, China

**Keywords:** Polygoni Multiflora Radix, processing, mononuclear anthraquinones, binuclear anthraquinones, hepatotoxicity, quality control

## Abstract

In recent years, the hepatotoxicity of Polygoni Multiflora Radix (PMR) has attracted increased research interest. Some studies suggest that anthraquinone may be the main hepatotoxic component. Most of the relevant studies have focused on the mononuclear anthraquinone component rather than binuclear anthraquinones. The hepatotoxicity of dinuclear anthraquinone (dianthrone) was investigated in a cell-based model. Next, a method for the determination of six free and total dianthonones in PMR and PMR Praeparata (PMRP) was established using ultra-high-performance liquid chromatography triple quadrupole mass spectrometry (UPLC-QQQ-MS/MS), which was then used to analyze the collected samples. The data show that four binuclear anthraquinone compounds were hepatotoxic and may be potential toxicity indicators for the safety evaluation of PMR and PMRP. Herein, we provide a theoretical basis for the improvement of PMRP quality standards.

## 1. Introduction

*Polygonum multiflorum* Thunb., the dried tuberous root of *Polygonum multiflorum*, has been shown to have detoxifying, anti-cancellation, anti-malarial, and laxative effects. It is widely used as a tonic herbal medicine because of its protective effects on the liver and kidneys, blood, tendons, and bones. Moreover, *Polygonum multiflorum* Thunb. can darken hair, as well as reduce turbidity and fat [1,2,3,4]. However, liver injury is the main adverse effect of the Polygoni Multiflora Radix medicine, of which case reports are common in China and abroad. Clinical studies have shown that adverse reactions, such as liver injury, are related to the dose, degree of specification, and course of treatment used. Most cases resolve once the drug treatment is stopgure Fiped; however, some severe cases can result in death. Basic studies on hepatotoxic substances indicate that anthraquinones may be involved in hepatotoxicity [5].

HepG2 cells were derived from the liver tissue of a 15-year-old white male with a well-differentiated hepatocellular carcinoma [6]. These cells were used to study the toxic effects of drugs in vitro because they preserve most of the metabolic functions of normal liver cells [7]. Furthermore, HepG2 cells have been used to evaluate the cytotoxic effects of sesquiterpene lactones, using the 3-(4,5-dimethylthiazol-2-yl)-2,5-diphenyltetrazolium bromide (MTT) assay [8]. These cells have also been used to investigate the toxic effects of extractions of *Frankenia laevis* L. (sea heath) [9].

Dianthrone belongs to binuclear anthraquinone and has been studied as a potential hepatotoxic compound. In previous studies, researchers have used an AB wild-type zebrafish model for hepatotoxicity evaluation, where the liver area, yolk sac area, and liver area grayscale were used as indicators [10]. The results indicated that both *cis*-emodin-emodin dianthrones and *trans*-emodin-emodin dianthrones were hepatotoxic compounds. In another study, 39 toxic components found in PMR were entered into toxicology prediction software; the data suggested that dianthrone may be highly toxic [11]. Other studies have also determined the components of four dianthrone glycosides and two free dianthrones by standard methods and found that these components decreased following processing. This followed the same trend as the observed change in toxicity, suggesting that dianthrone components may be potential quality control indicators for the prediction of PMR toxicity, as well as a quality control tool for PMR manufacture [12].

Herein, we evaluated the hepatotoxicity of four dianthrones in HepG2 cells and we established an assay to distinguish between six dianthrone components. Moreover, we performed content determination on different samples to give a reference for content limits and provide a theoretical basis for the refinement and improvement of the quality standard of PMRP.

## 2. Results and Discussion

### 2.1. Optimization of the Extraction Method

PMR (batch number: GDDQ202109) was used to optimize the extraction. In the analysis of free dianthrones, the type of extractant, volume of extractant, and ultrasound time were investigated (Figure 1)**.** The results showed that the extraction efficiency was highest when 50 mL of 50% (*v*/*v*) ethanol was used and extracted for 30 min, which was chosen as the final sample treatment method. For the determination of total dianthrones, the effects of the volume of dilute hydrochloric acid, hydrolysis temperature, hydrolysis time, and volume of extraction solvent were investigated (Figure 2). The extraction efficiency of the hydrolysis times of 30 min, 60 min, and 90 min was similar; to save sample-processing time and energy, 30 min was chosen as the extraction time. The final conditions that we decided on were 10 mL of dilute hydrochloric acid, added and hydrolyzed at 50 °C for 30 min. Extraction was performed twice, using 30 mL of ethyl acetate each time.

### 2.2. Optimization of UHPLC-QQQ-MS/MS Conditions

During the optimization of chromatographic conditions, the mobile phase composition was investigated first. Methanol-water and acetonitrile-water were compared, after which the addition of formic acid was examined, and it was found that the best separation was achieved with acetonitrile-0.1% formic acid. Secondly, the mobile phase gradient was optimized. For the quantitative analysis, one quantitative ion pair and one qualitative ion pair were selected for each compound. The cone and collision energy values were obtained via the automatic optimization of the instrument.

### 2.3. Method Validation

#### 2.3.1. Specificity

Here, 50% ethanol (*v*/*v*) was selected as the blank solution. The blank solution, the sample solution, and the standard solution were injected for analysis; the resulting chromatograms are shown in Figure 3. The blank solution had no interfering peaks at the corresponding positions of the test compounds. Furthermore, the compound peaks were well-separated, suggesting that the method is specific and can be used to study the compound constituents.

#### 2.3.2. Precision

In terms of free dianthrones, intra-day and inter-day precision were evaluated by means of the sample solution (batch number: GDDQ202109). The intra-day precision was tested in one day (*n* = 6) and the inter-day precision was tested on three consecutive days (*n* = 3). The peak areas of the six dianthrone components were recorded for six consecutive injections and three consecutive days. The RSDs for intra-day precision were 4.25%, 2.83%, 4.81%, 4.06%, 5.17%, and 5.99%, respectively, while the RSDs for inter-day precision were less than 22.13% (Table 1).

In terms of total dianthrone, intra-day and inter-day precision values were evaluated in the same way. The RSDs for intra-day precision were 1.56%, 1.36%, 1.20%, 1.04%, 1.49%, and 1.62%, respectively, while the RSDs for inter-day precision were less than 11.05% (see Table 2).

#### 2.3.3. Stability and Repeatability

The peak areas of the sample solution were measured at 0 h, 2 h, 4 h, 8 h, 10 h, 12 h, 18 h, and 24 h, and the RSDs were calculated. The results showed that the sample solution was stable within 24 h (Table 1 and Table 2). As to reproducibility, six sample solutions were prepared in parallel (*n* = 6), injected, and analyzed for their constituents, then the RSDs of each compound were calculated. Our data show that this method is highly reproducible (Table 1 and Table 2).

#### 2.3.4. Linearity Range, Limits of Detection (LODs), and Limits of Quantification (LOQs)

The standard solution was diluted stepwise to obtain a series of mixed standard solutions of different concentrations. The sample was injected, and then the peak areas were recorded. A standard curve was plotted using the concentration of the control on the x-axis and the peak area of the control on the y-axis, and linear regression analysis was performed (Table 3 and Table 4). The mixed standard solution was diluted and measured to determine the detection limit (S/N = 3) and quantification limit (S/N = 10) of each compound (Table 3 and Table 4).

#### 2.3.5. Recovery

We took 0.5 g of the sample powder (batch number: GDDQ202109) and mixed this in a 1:1 ratio with the standard solution. Next, we prepared 6 copies of the test solution in parallel, injected the sample solution and recorded the peak areas. Finally, we calculated the average recoveries and RSDs of the 6 compounds. The recoveries were calculated using the following equation: recovery (%) = (total amount detected – amount original)/amount spiked × 100%. The data show that this method is suitable for determining the six dianthrone components in these samples (Table 1 and Table 2).

### 2.4. Results of Quantitative Analysis

#### 2.4.1. Quantification of the Six Free Dianthrones in PMR

The results of the quantification of free dianthrone in PMR are shown in Appendix A. The value for total free dianthrones refers to the sum of the contents of the six compounds. Among the samples collected from different areas (PRM-01–PRM-57), the contents of samples 1–6 were in the ranges of 0.0813–76.8091, 0.0803–71.7398, 0.0643–63.8661, 0.0417–67.3944, 0.0640–113.3277, and 0.0424–73.5961 µg/g, respectively. The total contents of 1–6 ranged from 0.8469 to 380.3744 µg/g, with large differences up to three orders of magnitude. When the mean value of total free dianthrones of these samples was calculated, the highest value (353.1628 μg/g) was found in the Gaozhou production area in Guangdong Province, while the lowest value (3.8319 μg/g) was found in the Bijie production area in Guizhou province. The results of the content determination of each batch of herbs were higher than the instrumental detection limit, indicating that these six dianthrones are commonly found in PMR and that there is a risk of adverse reactions of liver damage from its use.

Among the samples collected from Guangdong, Deqing Province (PRM-58–PRM-97), the contents of samples 1–6 were in the ranges of 0.1153–8.4463, 0.0189–10.7348, 0.1033–8.5174, 0.0631–9.8430, 0.1990–44.4808, and 0.1017–16.7732 µg/g, respectively. The total contents of 1–6 ranged from 0.6013 to 96.7732 µg/g. PRM-58–PRM-87 were grown for different lengths of time. It was found that the total free dianthrone content decreased with time, indicating that PRM safety increased when dianthrone was used as an indicator of toxicity. This suggests that it may be reasonable to use PRM that is over 3 years old for medicinal purposes. In this experiment, the total free dianthrone content of 5-year-old PMR was higher than in 4-year-old PMR. This was likely an artifact caused by the higher content in one batch of samples. PRM-78–PRM-83, and PRM-88–PRM-97 were harvested in different seasons; a comparison of the mean values of these contents showed that the total free dianthrone content was the lowest in the samples harvested in summer and highest in the samples harvested in autumn for the same growth period and in the same production area. This may be consistent with the pattern of substance accumulation. Summer is a period of rapid growth with low content of each component, while autumn and winter are the seasons in which various components accumulate, leading to increased levels of each component.

#### 2.4.2. Quantification of the Six Free and Total Dianthrones in 90 Batches of PMRP

The results of the determination of the content of PRMP collected from herb markets and pharmaceutical enterprises are shown in Appendix A. To facilitate the comparison of the changes in the dianthrone content of PMR before and after processing, the results of the determination of samples collected from the same pharmaceutical enterprises are also presented in Appendix A. “Total dianthrone” indicates the content after hydrolysis (total dianthrone content = free dianthrone content + bound dianthrone content). Among the PRMP samples, the contents of 1, 2, 3, 4, 5, and 6 before hydrolysis were in the ranges of 0.0000–23.4415, 0.0000–23.6801, 0.0000–17.4026, 0.0000–19.2068, 0.0000–48.4669, and 0.0000–35.0726 µg/g, respectively. The total contents of 1–6 ranged from 0.0000 to 135.2258 µg/g. The contents of 1, 2, 3, 4, 5, and 6 after hydrolysis were in the ranges of 0.0000–168.247, 0.0000–183.6098, 0.0000–96.7821, 0.0000–115.6580, 0.0000–171.1238, and 0.0000–83.1678 µg/g, respectively. The total contents of 1–6 ranged from 0.0000 to 499.7663 µg/g. The results showed that the content measured after hydrolysis was much higher than before hydrolysis. After processing, the dianthrone content decreased; the longer the processing time, the more obvious this decrease.

#### 2.4.3. Quantification of the Six Dianthrones in PMRP, Processed by Different Methods

The results of the determination of the content of PRMP are shown in Appendix A. Among the PRMP samples, the contents of 1, 2, 3, 4, 5, and 6 before hydrolysis were in the ranges of 0.0000–64.0366, 0.0000–61.9686, 0.0000–63.8661, 0.0000–67.3944, 0.0000–51.1396, and 0.0000–32.6669 µg/g, respectively. The total contents of 1–6 ranged from 0.0000 to 162.9507 µg/g. The contents of 1, 2, 3, 4, 5, and 6 after hydrolysis were in the ranges of 0.0000–194.2306, 0.0000–215.0783, 0.0000–157.6543, 0.0000–194.0987, 0.0000–231.6045, and 0.0000–156.7588 µg/g, respectively. The total contents of 1–6 ranged from 0.0251 to 1149.425 µg/g. Data from the content determination of the dianthrones in PMRP prepared by different methods showed that the total dianthrone content of the four processes showed a decreasing trend with increasing processing time. The decreases were greater in the first few hours of processing, and the changes seemed to level off with time. Among them, the steaming process (PMRP-Q_2h_–PMRP-Q_48h_) and the black bean juice steaming process (PMRP-H_2h_–PMRP-H_48h_) used the same batch of PMR (PMR-12), and the six dianthrone contents were nearly the same at the different processing times during processing, indicating that the reduction in toxicity in the two processes was comparable when using this as an indicator. The content of six dianthrones in the nine-steaming and nine-making samples (PMRP-JZJZ_1_) was close to that in the stewed sample (PMRP-DZ), both of which had the same processing time and the same processing materials, indicating that the two processes had similar effects on the dianthrone content. Comparing the four processes, using the total dianthrone content as the index, the end-product contents of the nine steaming and nine stewing processes decreased the most and were considered the best process methods.

### 2.5. Cytotoxic Effects of Dianthrone Exposure in HepG2 Cells

The toxicity results of the four dianthrone compounds in human HepG2 cells at concentrations of 10 μM-40 μM for 48 h are shown in Table 5. The *trans*-emodin-physcion dianthrones and *cis*-emodin-physcion dianthrones showed significant hepatotoxicity. Further, *cis*-emodin-physcion dianthrones were significantly more toxic than *trans*-emodin-physcion dianthrones. The *trans*-physcion-physcion dianthrones and *cis*-physcion-physcion dianthrones did not exhibit significant hepatotoxicity under the conditions tested.

### 2.6. Discussion

Firstly, based on the results of the cellular experiments, both *trans*-emodin-physcion dianthrones and *cis*-emodin-physcion dianthrones significantly decreased the cell OD value and cell survival, indicating that these two binuclear anthraquinone components have hepatotoxic effects and are potential hepatotoxic substances of PMR. This provides proof-of-concept for using dianthrone as a toxicity indicator in the quality control of PMR production. Nonetheless, its toxicity should be further explored in in vivo models, such as in zebrafish and mice. The absence of significant positive reactions in the *trans*-physcion-physcion dianthrones and *cis*-physcion-physcion dianthrones groups does not necessarily suggest that they are safe. Rather, their toxicity should also be tested in other models.

Secondly, according to other hepatotoxicity studies in PMR, anthraquinones are the hepatotoxic components of PMR. We found that the toxicity of PMR decreased after processing. The results of content determination showed that the content of mononuclear anthraquinones, such as emodin, increased after processing [13], while the content of binuclear anthraquinones, such as dianthrone, decreased or disappeared, suggesting that mononuclear anthraquinones may not be the main hepatotoxic substances and that dianthrone components may also be potential hepatotoxic components. According to previous studies, dianthrones and other dinuclear anthraquinones are hepatotoxic in both cellular and zebrafish models. These studies also suggest that dianthrones and other dinuclear anthraquinone components should be studied further. The method established in this paper for the determination of free dianthrone content is easy and accurate and can be used for PMR and PRMP quality control. The dianthrones were detected in all collected PMR samples, which suggests that PMR may cause liver damage. The method established in this paper for the determination of dianthrone content after hydrolysis allows for the determination of the sum of free and bound dianthrone contents. Our content determination data showed that a significant portion of dianthrone exists as bound dianthrone. Because the hydrolysis conditions in our samples mimic the environment in the human stomach, and because we presumed that bound dianthrone could be converted into free dianthrone in the stomach, we considered it more reasonable to use the sum of the content of the six compounds after hydrolysis as an indicator of PMR hepatotoxicity.

Thirdly, the processing method has a significant impact on the quality of PMRP, of which the processing time, auxiliary materials, processing methods, and drying method are the process parameters of traditional Chinese medicine. The Chinese Pharmacopoeia and others do not specify these parameters; they only qualify the properties of the end product, which makes quality control difficult. The experimentally collected samples of PMRP from pharmaceutical enterprises have clear processing methods, which can be used for PMR processing methods and quality control studies. Most companies used black bean juice as a processing excipient, presumably because the addition of black bean juice changes the color more quickly and makes it easier to meet the requirements of the relevant standards. Moreover, studies have shown that black bean juice can enhance the efficacy of PMRP [14,15]. The sample with the longest processing time that was collected from each company was used as a reasonably concocted, good quality, prepared PMRP, and was found to contain no more than 9.1257 μg/g of total dianthrones. We concluded that the total dianthrone content in prepared PMRP should not exceed this limit. This limit was used to evaluate the 30 batches of PMRP collected from medicinal markets. A failure rate of more than 50%, indicating that the PMRP samples varied in quality, is suggestive of an increased risk of adverse reactions. Further investigations are required to set the relevant toxic component limit standards, improve quality standards, and maintain consumer safety.

## 3. Materials and Methods

### 3.1. Plant Materials and Reagents

One hundred and thirteen batches of samples were collected (Appendix A). Fifty-seven batches of PMR (PMR-01–PMR57) were collected from different provinces in China (Appendix A). Forty batches of PMR (PMR-58–PMR-97) were collected from Deqing, Guangdong Province, in China. Samples PMR-58 to PMR-87 were picked in the autumn; samples PMR-58–PMR-62, PMR-63–PMR-67, PMR-68–PMR-72, PMR-73–PMR-77, PMR-78–PMR-82, and PMR-83–PMR-87 were collected 1-, 2-, 3-, 4-, 5-, and 6 years, respectively, before this work commenced. In addition, the PMR-88–PMR-92 samples were harvested in spring, while the PMR-93–PMR-97 samples were harvested in summer, 5 years prior to this study. Fifteen batches of PMR (PMR-98–PMR113) were collected from different pharmaceutical companies. The processed samples are shown in Appendix A. PMR and PMRP samples were authenticated by Associate Professor Jian-Bo Yang (Research and Inspection Center of TCM and Ethno-medicine, National Institutes for Food and Drug Control, State Food and Drug Administration) in accordance with the Chinese Pharmacopoeia (edition 2020, volume 1).

Seventy-six batches of PMR (PMR-01–PMR-76) were collected from herb markets and pharmaceutical companies.

Four batches of PMRP were made from PMR using the method from the Chinese Pharmacopoeia (2020 edition) and traditional methods. Eighty-six batches of PMR (PMRP-01–PMRP-86) were collected from different provinces of China, as shown in Appendix A.

Both the water-steaming method and black bean juice-steaming method are traditional concoction techniques recorded in the ancient texts and included in the Chinese Pharmacopoeia. The manufacture of two batches of PMRP was carried out as follows: approximately 1 kg of PRM (batch number: GDGZ202003) was taken and added to 670 mL of ultrapure water or soya-bean milk and mixed well, before being left to absorb the liquid overnight. Subsequently, this mixture was placed on a wooden steamer and heated over water for 2 h, 4 h, 8 h, 12 h, 18 h, 24 h, 30 h, 36 h, and 48 h. The samples were then left to dry in the sun to obtain two types of PRMP samples: steamed (labeled PRMP-Q_2h_, Q_4h_, Q_8h_, Q_12h_, Q_18h_, Q_24h_, Q_30h_, Q_36h_, and Q_48h_, respectively) and mixed with black bean juice (labeled PRMP-H_2h_, H_4h_, H_8h_, H_12h_, H_18h_, H_24h_, H_30h_, H_36h_, and H_48h_, respectively). Black bean juice was prepared in accordance with the Chinese Pharmacopoeia (2020 edition).

The nine-steam-nine-bask processing method was as follows: approximately 1 kg of PRM (batch number: GDDQ2020) was taken and added to 800 mL of ultrapure water and black bean. This mixture was then left to incubate overnight. At the same time, 4 kg of black beans were added to 2 L of pure water and left to moisten overnight. Subsequently, the black beans were placed in the steam drawer, then PRM was put onto the black beans, followed by water steaming for 9h. We then took out the samples, discarded the black beans, dried the mixture, obtained a steam system sample, and repeated the peocess nine times to obtain the other samples (labeled PRMP-JZJZ_1_, JZJZ_2_, JZJZ_3_, JZJZ_4_, JZJZ_5_, JZJZ_6_, JZJZ_7_, JZJZ_8_, JZJZ_9_).

The stewing method was as follows: about 1 kg of PRM (batch number: GDDQ2020) was added to black bean juice and placed in a suitable non-iron-based container, wherein the mixture was stewed until the juice was absorbed (8 h). Next, the juice was removed and dried, to obtain stewed samples (labeled PRMP-DZ).

High-performance liquid chromatography-grade acetonitrile and methanol were purchased from Merck (Muskegon, MI, USA). Water was purified with a Milli-Q water purification system (Millipore, Billerica, MA, USA). Carboxymethyl cellulose-Na was purchased from the Xilong Chemical Co., Ltd. (Shantou, China). Analytical grade ethanol was purchased from Sinopharm Chemical Regent Co., Ltd. (Shanghai, China), and formic acid (LC-MS grade) was purchased from the Sigma-Aldrich Corporation (St. Louis, MO, USA). *Trans*-emodin-emodin dianthrones (Compound 1, No. JJ9735-TS20C127-DT01, purity ≥ 99.5%) and *cis*-emodin-emodin dianthrones (Compound 2, No. JJ9735-TS20C127-DT02, purity ≥ 99.5%), *trans*-emodin-physcion dianthrones (Compound 3, No. JJ9735-TS20C127-DT03, purity ≥ 97.8%) and *cis*-emodin-physcion dianthrones (Compound 4, No. JJ9735-TS20C127-DT04, purity ≥ 99.2%), and *trans*-physcion-physcion dianthrones (Compound 5, No. JJ9735-TS20C127-DT05, purity ≥ 99.8%) and *cis*-physcion-physcion dianthrones (Compound 6, No. JJ9735-TS20C127-DT06, purity ≥ 96.9%), were purchased from the Shanghai Standard Technology Co., Ltd. (Shanghai, China). The structures of the six dianthrones are shown in Figure 4.

### 3.2. Instruments and Conditions

Two quantitative assays were performed with an Acquity™ UPLC I-class system equipped with a photo-diode array and a Waters Xevo TQ-S micro triple quadrupole mass spectrometer detector (Waters Corp, Milford, MA, USA). Chromatographic separation was carried out at 30 °C on an ACQUITY UPLC ® CSHTM C18 (2.1 × 100 mm; 1.7 μm). The mobile gradient phase was composed of acetonitrile (A) and water containing 0.1% (*v*/*v*) formic acid (B) at a flow rate of 0.3 mL/min. The gradient was programmed as follows: 0–4 min, linear change from 45% to 60% A; 4–14 min, maintaining 60% A; 14–16 min, linear change to 70% A; 16–21 min, maintaining 70% A; 21–24 min, maintaining 95% A; 24–30 min, maintaining 45% A. The column temperature was maintained at 30 °C. The injection volume was set to 2.0 μL. Regarding MS conditions, the multiple reaction monitoring (MRM) mode was chosen. Nitrogen was used as the both nebulizer and collision gas. Electrospray ionization (ESI) was selected as the ionization source and the negative ionization mode was set as the detection mode. Ion source parameters were as follows: capillary voltage, 3.0 kV for the negative mode; source temperature, 150 °C; desolvation temperature, 450 °C; desolvation gas flow rate, 1000 L/h; cone gas (nitrogen) flow rate, 150 L/h. The dianthrones parameters are detailed in Table 6.

### 3.3. Preparation of Solutions

#### 3.3.1. Preparation of Standard Solutions

Stock solutions for *trans*-emodin-emodin dianthrones and *cis*-emodin-emodin dianthrones, *trans*-emodin-physcion dianthrones, *cis*-emodin-physcion dianthrones, *trans*-physcion-physcion dianthrones, and *cis*-physcion-physcion dianthrones were made to 80.1 μg/mL, 80.6 μg/mL, 80.1 μg/mL, 80.2 μg/mL, 7.2 μg/mL, and 8.1 μg/mL, respectively. The stock solutions were then diluted to 769.0 ng/mL, 344.6 ng/mL, 608.8 ng/mL, 545.4 ng/mL, 594.5 ng/mL, and 259.2 ng/mL, respectively, before being filtered with a 0.22 μm membrane for UHPLC.

#### 3.3.2. Preparation of Sample Solution (Free Dianthrones)

Approximately 1 g of PMR or PMRP powder (filtered through a no. 4 sieve) was weighed and placed in a 50 mL conical flask with a stopper, then 50 mL of 50% (*v*/*v*) ethanol was added, sonicated for 30 min (power 300 W, frequency 40 kHz), mixed well, and then weighed again after cooling. Next, the lost weight was made up with 50% (*v*/*v*) ethanol before filtration to obtain the sample solution. The sample solution was again filtered through a 0.22-μm filter membrane before analysis.

#### 3.3.3. Preparation of Sample Solution (Total Dianthrones)

Approximately 1 g of PMR or PMRP powder (filtered through a no. 4 sieve) was weighed and placed in a 50 mL conical flask with a stopper, then 50 mL of 50% (*v*/*v*) ethanol was added, sonicated for 30 min (power 300 W, frequency 40 kHz), mixed well, and weighed again after it had cooled down. The lost weight was again made up with 50% (*v*/*v*) ethanol, before being filtered. Subsequently, 10 mL of the resulting filtrate was taken and recycled to dry. The remaining residue was dissolved in dilute hydrochloric acid solution (10 mL) and placed in a water bath at 50 °C for 30 min. (The purpose of adding the hydrochloric acid is to break the glycosidic bond and convert the bound dianthrone to free dianthrone.) The sample was then removed, cooled, and decanted into a separatory funnel. The container was then washed with a small amount of ethyl acetate, and the residue was extracted twice, using 30 mL of ethyl acetate each time. The sample was then separated, and the ethyl acetate was collected and recycled to dry. Next, 50% (*v*/*v*) ethanol was added to dissolve the residue, the solvent was then transferred to a 25 mL bottle, and 50% (*v*/*v*) ethanol was added to the scale before shaking well. The sample solution was filtered through a 0.22-μm filter membrane before analysis. Total dianthrone content = free dianthrone content (genin) + bound dianthrone content (glycosides).

### 3.4. Cytotoxic Effects of Dianthrone Exposure in HepG2 Cells

Human hepatocellular carcinoma HepG2 cells were used because they reflect the characteristics of normal human hepatocytes. These cells were grown in Dulbecco’s modified Eagle medium (DMEM) containing 10% (*v*/*v*) fetal bovine serum (containing 100 U/mL of penicillin and 100 µg/mL of streptomycin) at 37 °C, 5% CO_2_, and saturated humidity. The transfections were digested with a solution containing 0.25% trypsin and 0.02% EDTA. The reference substance of *trans*-emodin-physcion dianthrones, *cis*-emodin-physcion dianthrones, *trans*-physcion-physcion dianthrones, and *cis*-physcion-physcion dianthrones were dissolved with DMSO to a concentration of 20 mM and then diluted to 10 μM, 20 μM, 40 μM, and 80 μM. The HepG2 cells were seeded in 96-well cell culture plates for the subsequent MTT assays. After 24 h, different concentrations of the test substances were added. A solvent control group was set up, with 3–4 parallel wells for each drug concentration. After 48 h of drug treatment, the culture medium was discarded and 100 μL of MTT (0.5 mg/mL) solution was added to the remaining adhered cells in each well; the culture was continued for an additional 4 h. The MTT solution was then discarded and 150 μL of DMSO was added to each well before the mixture was shaken with an oscillator. Absorbance was then measured at 570 nm using an enzyme standard meter. The cell survival rate was calculated using the following formula: cell survival rate (%) = (mean OD value of administered cells/mean OD value of solvent control cells) × 100%.

## 4. Conclusions

In this study, it was shown that anthraquinones may be the source of the hepatotoxicity of PMR. The results of the content determination showed that the content of binucleated anthraquinones was stable in PMR and decreased or disappeared after processing. When combined with the pattern of toxicity reduction in PMR after processing, it is suggested that the content of binucleated anthraquinones can be used as an index for a toxicity evaluation of PMR and as an aid to judging the degree of processing needed for PMRP, which can be used to improve and enhance the quality standard of commercial PMRP.

## Figures and Tables

**Figure 1 molecules-27-06760-f001:**
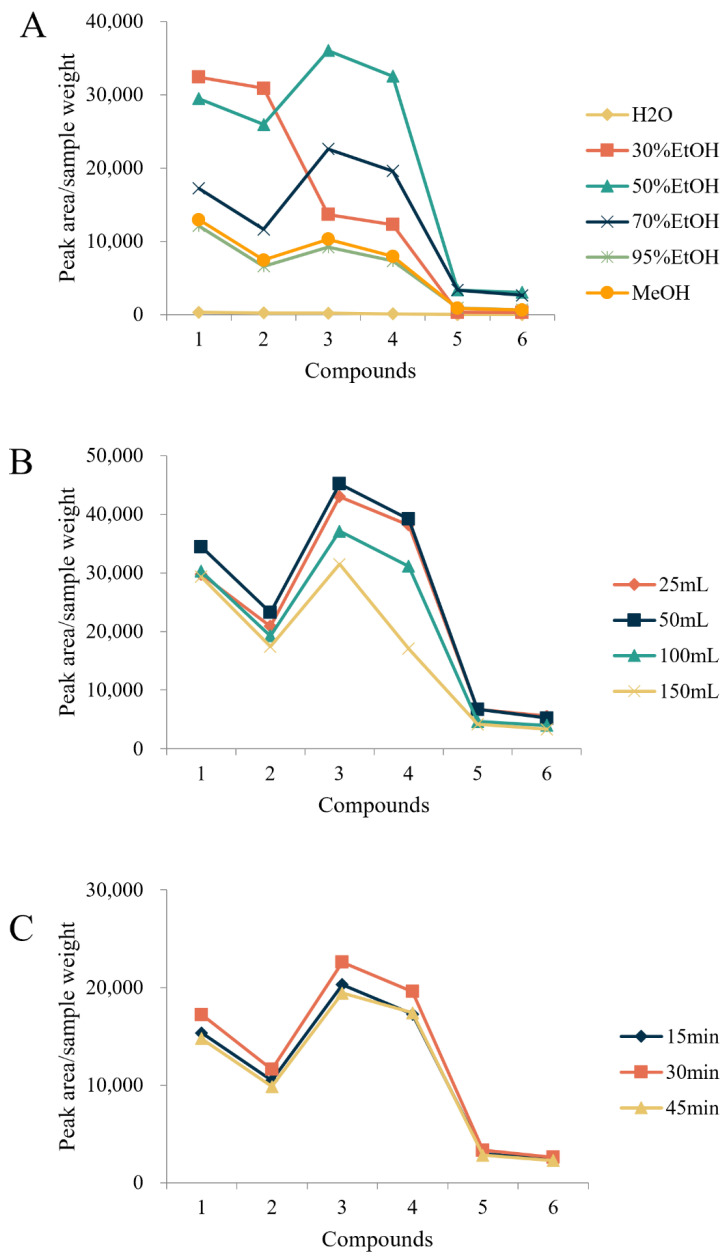
Optimization of the different parameters of the free dianthrones sample solution: (**A**) types of extractants, (**B**) volume of extractant, and (**C**) ultrasound time. Compounds: *trans*-emodin-emodin dianthrones (1), *cis*-emodin-emodin dianthrones (2), *trans*-emodin-physcion dianthrones (3), *cis*-emodin-physcion dianthrones (4), *trans*-physcion-physcion dianthrones (5), and *cis*-physcion-physcion (6).

**Figure 2 molecules-27-06760-f002:**
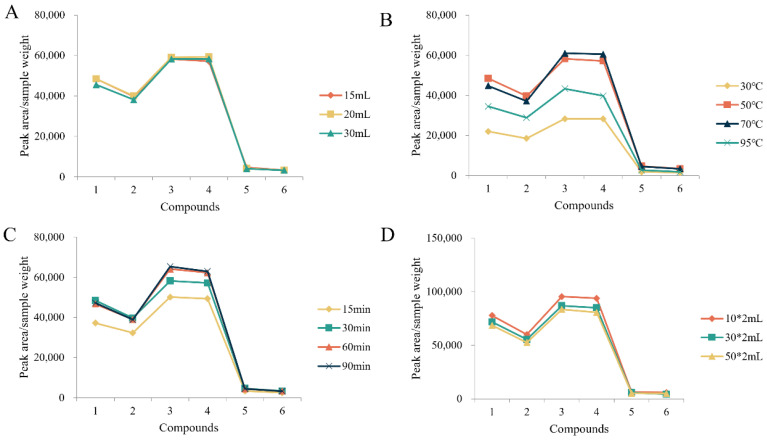
Optimization of different parameters of total dianthrones sample solution: (**A**) volume of dilute hydrochloric acid, (**B**) hydrolysis temperature, (**C**) hydrolysis time, and (**D**) volume of extraction solvent. Compounds: *Trans*-emodin-emodin dianthrones (1), *cis*-emodin-emodin dianthrones (2), *trans*-emodin-physcion dianthrones (3), *cis*-emodin-physcion dianthrones (4), *trans*-physcion-physcion dianthrones (5) and *cis*-physcion-physcion (6).

**Figure 3 molecules-27-06760-f003:**
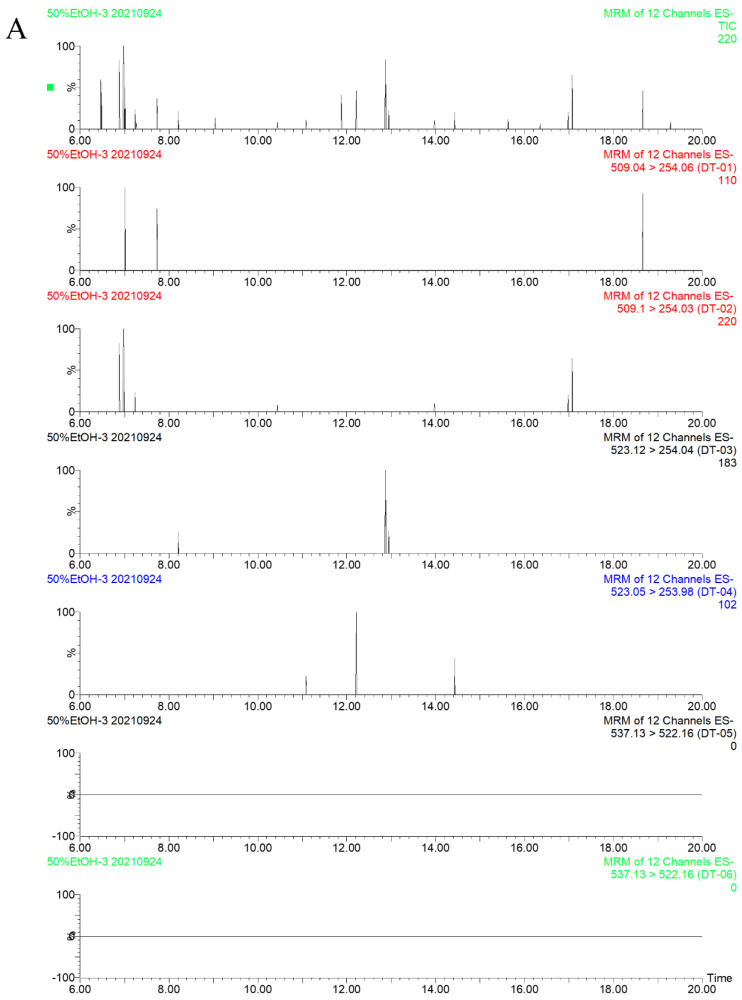
MS chromatograms of the blank solution (**A**), standard solution (**B**), and sample (**C**). *Trans*-emodin-emodin dianthrones (1), *cis*-emodin-emodin dianthrones (2), *trans*-emodin-physcion dianthrones (3), *cis*-emodin-physcion dianthrones (4), *trans*-physcion-physcion dianthrones (5) and *cis*-physcion-physcion (6).

**Figure 4 molecules-27-06760-f004:**
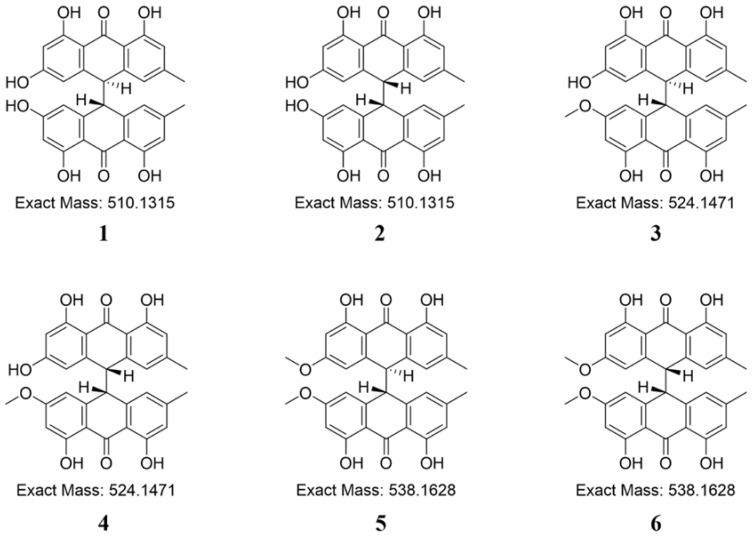
The chemical structure of six dianthrones (numbered 1–6). *Trans*-emodin-emodin dianthrones (1), *cis*-emodin-emodin dianthrones (2), *trans*-emodin-physcion dianthrones (3), *cis*-emodin-physcion dianthrones (4), *trans*-physcion-physcion dianthrones (5) and *cis*-physcion-physcion dianthrones (6).

**Table 1 molecules-27-06760-t001:** The stability, repeatability, precision, and recovery of 6 compounds in the detection of free dianthrones.

No. *	Stability RSD (%)	Repeatability RSD (%) (*n* = 6)	Precision		Recovery (*n* = 6)						
			Intra-Day RSD (%) (*n* = 6)	Inter-Day RSD (%) (*n* = 3)	Sample (g)	Original (μg) ^△^	Spiked (μg) ^1^	Found (μg) ^2^	Recovery (%)	Average Recovery (%)	RSD (%)
1	6.39	2.58	4.25	17.69	0.5002	2.0403	1.4060	3.4049	97.06	91.12	5.12
					0.5004	2.0411	1.4060	3.3355	92.06		
					0.5005	2.0415	1.4060	3.3790	95.13		
					0.5001	2.0399	1.4060	3.3147	90.68		
					0.5004	2.0411	1.4060	3.2627	86.89		
					0.5005	2.0415	1.4060	3.2353	84.91		
2	4.70	4.27	2.83	15.17	0.5002	0.8368	0.5192	1.4506	118.23	110.08	5.43
					0.5004	0.8371	0.5192	1.4078	109.92		
					0.5005	0.8373	0.5192	1.4376	115.62		
					0.5001	0.8366	0.5192	1.3929	107.15		
					0.5004	0.8371	0.5192	1.3660	101.87		
					0.5005	0.8373	0.5192	1.3965	107.71		
3	7.94	2.66	4.81	22.13	0.5002	1.4312	0.9562	2.4394	105.44	93.54	8.55
					0.5004	1.4318	0.9562	2.3309	94.04		
					0.5005	1.4321	0.9562	2.3881	99.99		
					0.5001	1.4309	0.9562	2.2962	90.50		
					0.5004	1.4318	0.9562	2.2406	84.59		
					0.5005	1.4321	0.9562	2.2608	86.67		
4	6.42	4.26	4.06	18.72	0.5002	1.3005	0.7841	2.1913	113.61	102.11	8.02
					0.5004	1.3010	0.7841	2.0904	100.68		
					0.5005	1.3013	0.7841	2.1634	109.95		
					0.5001	1.3003	0.7841	2.0859	100.20		
					0.5004	1.3010	0.7841	2.0231	92.08		
					0.5005	1.3013	0.7841	2.0551	96.14		
5	5.60	5.06	5.17	15.28	0.5005	1.8202	0.9862	2.6891	88.11	85.93	7.20
					0.5002	1.8191	0.9862	2.7313	92.50		
					0.5004	1.8198	0.9862	2.6423	83.40		
					0.5005	1.8202	0.9862	2.7235	91.60		
					0.5001	1.8187	0.9862	2.5667	75.84		
					0.5005	1.8202	0.9862	2.6497	84.12		
6	7.94	4.66	5.99	16.80	0.5002	1.3005	0.7841	2.1913	113.61	102.11	8.02
					0.5004	1.3010	0.7841	2.0904	100.68		
					0.5005	1.3013	0.7841	2.1634	109.95		
					0.5001	1.3003	0.7841	2.0859	100.20		
					0.5004	1.3010	0.7841	2.0231	92.08		
					0.5005	1.3013	0.7841	2.0551	96.14		

* Compounds: *trans*-emodin-emodin dianthrones (No. 1), *cis*-emodin-emodin dianthrones (No. 2), *trans*-emodin-physcion dianthrones (No. 3), *cis*-emodin-physcion dianthrones (No. 4), *trans*-physcion-physcion dianthrones (No. 5) and *cis*-physcion-physcion (No. 6). **^△^** Amount originally in the sample. ^1^ Amount of compound that was added. ^2^ Total amount detected.

**Table 2 molecules-27-06760-t002:** The stability, repeatability, precision, and recovery of 6 compounds in the detection of total dianthrones.

No. *	Stability RSD (%)	Repeatability RSD (%) (*n* = 6)	Precision		Recovery (*n* = 6)						
			Intra-Day RSD (%) (*n* = 6)	Inter-Day RSD (%) (*n* = 3)	Sample (g)	Original (μg) ^△^	Spiked (μg) ^1^	Found (μg) ^2^	Recovery (%)	Average Recovery (%)	RSD (%)
1	6.05	4.44	1.56	8.96	0.5004	9.1247	8.0449	16.2095	88.07	81.10	7.37
					0.5000	9.1174	8.0449	16.1979	88.01		
					0.5000	9.1174	8.0449	15.6875	81.67		
					0.5003	9.1229	8.0449	15.4832	79.06		
					0.5004	9.1247	8.0449	15.1956	75.46		
					0.5004	9.1247	8.0449	15.1027	74.31		
2	6.98	5.43	1.36	9.86	0.5004	8.7546	6.7196	15.1793	95.61	85.40	11.08
					0.5000	8.7476	6.7196	15.1753	95.66		
					0.5000	8.7476	6.7196	14.6383	87.67		
					0.5003	8.7528	6.7196	14.4006	84.05		
					0.5004	8.7546	6.7196	13.8468	75.78		
					0.5004	8.7546	6.7196	13.7023	73.63		
3	6.13	4.82	1.20	9.44	0.5004	7.5397	6.6519	12.9933	81.99	76.73	6.29
					0.5000	7.5337	6.6519	12.9775	81.84		
					0.5000	7.5337	6.6519	12.5910	76.03		
					0.5003	7.5382	6.6519	12.7026	77.64		
					0.5004	7.5397	6.6519	12.1783	69.73		
					0.5004	7.5397	6.6519	12.4074	73.18		
4	5.32	4.46	1.04	8.59	0.5004	9.6078	8.7507	16.8310	82.54	77.37	6.78
					0.5000	9.6001	8.7507	16.9669	84.19		
					0.5000	9.6001	8.7507	16.2378	75.85		
					0.5003	9.6058	8.7507	16.4300	77.98		
					0.5004	9.6078	8.7507	15.8746	71.62		
					0.5004	9.6078	8.7507	15.9094	72.01		
5	6.78	3.41	1.49	10.41	0.5004	8.2760	10.6095	19.6722	107.41	105.99	7.45
					0.5000	8.2694	10.6095	20.6117	116.33		
					0.5000	8.2694	10.6095	20.0914	111.43		
					0.5003	8.2744	10.6095	19.3897	104.77		
					0.5004	8.2760	10.6095	18.1766	93.32		
					0.5004	8.2760	10.6095	18.5247	102.70		
6	6.56	6.12	1.62	11.05	0.5004	6.1007	5.8564	11.3627	89.85	83.24	8.06
					0.5000	6.0958	5.8564	11.4982	92.25		
					0.5000	6.0958	5.8564	10.6751	78.19		
					0.5003	6.0995	5.8564	11.0595	84.69		
					0.5004	6.1007	5.8564	10.6269	77.29		
					0.5004	6.1007	5.8564	10.6198	77.16		

* Compounds: *trans*-emodin-emodin dianthrones (No. 1), *cis*-emodin-emodin dianthrones (No. 2), *trans*-emodin-physcion dianthrones (No. 3), *cis*-emodin-physcion dianthrones (No. 4), *trans*-physcion-physcion dianthrones (No. 5) and *cis*-physcion-physcion (No.6). ^△^ Amount originally in the sample. ^1^ Amount of compound that was added. ^2^ Total amount detected.

**Table 3 molecules-27-06760-t003:** Regression equation, range, LOD, and LOQ of the six compounds in the detection of free dianthrones.

No.	Compounds	Regression Equation	R^2^	Range (ng/mL)	LOD (ng/mL)	LOQ (ng/mL)
1	*trans*-emodin-emodin dianthrones	y = 524.18x + 3553.5	0.9941	1.20–306.04	0.60	1.20
2	cis-emodin-emodin dianthrones	y = 470.23x + 2282.8	0.9949	0.55–281.16	0.27	0.55
3	*trans*-emodin-physcion dianthrones	y = 699.51x + 4752.4	0.9939	0.93–357.22	0.47	0.93
4	*cis*-emodin-physcion dianthrones	y = 531.68x + 4300.9	0.9939	0.85–432.80	0.42	0.85
5	*trans*-physcion-physcion dianthrones	y = 44.934x + 37.402	0.9981	0.93–475.60	0.46	0.93
6	*cis*-physcion-physcion dianthrones	y = 72.981x + 121.73	0.9983	0.39–200.93	0.20	0.39

**Table 4 molecules-27-06760-t004:** Regression equation, range, LOD, and LOQ of the six compounds in the detection of total dianthrones.

No.	Compounds	Regression Equation	R^2^	Range (ng/mL)	LOD (ng/mL)	LOQ (ng/mL)
1	*trans*-emodin-emodin dianthrones	y = 437.66x + 3967.5	0.9911	0.83–423.36	0.41	0.83
2	cis-emodin-emodin dianthrones	y = 363.94x + 1279.5	0.9971	0.66–337.39	0.33	0.66
3	*trans*-emodin-physcion dianthrones	y = 642.03x + 3708.2	0.9926	0.69–350.65	0.34	0.69
4	*cis*-emodin-physcion dianthrones	y = 477.46x + 3564	0.9948	0.90–458.26	0.45	0.9
5	*trans*-physcion-physcion dianthrones	y = 37.473x – 2.0684	0.9997	0.84–429.20	0.42	0.84
6	*cis*-physcion-physcion dianthrones	y = 45.271x + 8.0834	0.9999	0.45–230.69	0.23	0.45

**Table 5 molecules-27-06760-t005:** Cytotoxic effects of dianthrone exposure in HepG2 cells.

No.	Compounds	Concentration (μM)	OD Value	Cell Survival Rate (%)
0	Control	-	1.91 ± 0.10	100.00
1	*trans*-emodin-physcion dianthrones	10	2.01 ± 0.08	105.0
		20	2.40 ± 0.05	125.56
		40	1.87 ± 0.04	97.80
		80	1.30 ± 0.05 ***	67.95
2	*cis*-emodin-physcion dianthrones	10	1.13 ± 0.06 ***	59.41
		20	1.09 ± 0.08 ***	57.02
		40	0.72 ± 0.05 ***	37.68
		80	1.14 ± 0.04 ***	59.96
3	*trans*-physcion-physcion dianthrones	10	1.77 ± 0.05	92.71
		20	1.72 ± 0.04	90.28
		40	1.71 ± 0.07 *	89.82
		80	1.81 ± 0.07	95.13
4	*cis*-physcion-physcion dianthrones	10	1.64 ± 0.05 *	86.09
		20	1.71 ± 0.02 *	89.46
		40	1.63 ± 0.03 **	85.58
		80	1.83 ± 0.10	95.68

* *p* < 0.05, ** *p* < 0.11, *** *p* < 0.001, compared to the Control group.

**Table 6 molecules-27-06760-t006:** Parameters of 6 dianthrones in multiple reaction monitoring (MRM) analysis.

No.	Compounds	Retention Times (RT, min)	Precursor Ion (m/z)	Quantitative Ion (m/z)	Qualitative Ion (m/z)	Cone (V)	Collision Energy (CE, V)	Ion Mode
1	*trans*-emodin-emodin dianthrones	6.96	509.0	254.1	225.1	24	20, 70	(–)ESI
2	*cis*-emodin-modin dianthrones	7.70	509.1	254.0	225.1	24	22, 68	(–)ESI
3	*trans*-emodin-physcion dianthrones	12.34	523.1	254.0	225.1	60	20, 60	(–)ESI
4	*cis*-emodin-physcion dianthrones	12.96	523.05	254.0	225.1	60	20, 76	(–)ESI
5	*trans*-physcion-physcion dianthrones	19.13	537.1	522.2	268.1	54	32, 28	(–)ESI
6	*cis*-physcion-physcion dianthrones	19.56	537.1	522.2	268.1	54	32, 28	(–)ESI

## Data Availability

All data can be found in this manuscript.

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
