# Peer review of "Exploratory Quality Control Study for Polygonum multiflorum Thunb. Using Dinuclear Anthraquinones with Potential Hepatotoxicity"

_molecules, 2022, doi:10.3390/molecules27196760_

Round 1
Reviewer 1 Report
Review of the manuscript entitled: "Exploratory quality control study for Polygonum multiflorum Thunb. using dinuclear anthraquinones with potential hepatotoxicity".
The manuscript presents a broad investigation of the content of six anthraquinones in Polygonum multiflorum herb and the formulations containing this plant with UPLC-QQQ-MS. The effect of geographic plant origin, harvest time, and mode of extract preparation were examined. Additionally, the toxicity of anthraquinones toward human hepatocellular carcinoma (HepG2) was evaluated in vivo. The authors have chosen six compounds to examine. They were simultaneously dianthrones and trans and cis isomers (emodin-emodin, emodin-physcion and physcion-physcion). They were investigated with UPLC-QQQ-MS chromatography gradient elution. Each step of the investigation was optimized. Chromatographic research was validated as the base of Polygonum multiflorum formulation quality control.
The reviewer's questions and suggestions:
Not all abbreviations used in the manuscript are not explained (e.g., MRM, DMEM, MTT..).
The compounds in Figures 2 and 3 are presented only as numbers, and the reader should guess that the numbers are the same as in Figure 1 or Table 3. So, the captures under Figures 2 and 3 should contain such information.
The reader unfamiliar with Chinese Traditional Medicine should find in the text the information that He Shou Wo is the traditional name for Polygonum multiflorum preparation. Also, it should be explained why the black-bean juice-steaming method is used (the reviewer assumes that it is the Traditional Chinese mode of preparing plant extracts).
Lines 45-48 are not apparent to the reader.
The manuscript contains a vast amount of data. Thus, it is easy to make a mistake.
Line 237 contains RSD intraday data from Table 4, while it should be from Table 5.
There is no discussion on the interday RSD results from table 4.
The results for trans physcion-physcion dianthrones (5) regarding the compound range in Table 6 are wrongly presented.
The result in line 314 regarding compound 5 is wrongly presented, too (range is 0.000048.4669 ?).
Authors could also indicate which compound is the main component of the preparations presented in Table 9? (According to the reviewer, it is compound 5). This table also contains several interesting pieces of information. Thus, it is evident that the anthraquinones content increases after hydrolysis, but why in the sample PRMP-03 and PRMP-05 it decreases? The decrease of the anthraquinones content after the hydrolysis (b) also is observed for the samples PMP-100 and PMP-101. What is the reason for such behaviour?
How to explain the significant increase of the compounds 1-6 after the hydrolysis (b line) in samples PRM-15 (50 times higher) or samples PRMP-28, PMP-112, and PMP-113?
How to explain that samples PRMP-54 and PRMP-55 contain the same amount of anthraquinones before (a) and after the hydrolysis (b)? All figures are the same.
The data presented in Table 10 should be better explained in the text. It is hard to understand differences in the sample preparation based on the sample abbreviation. The total content of the anthraquinones after the hydrolysis in Table 10 is wrongly calculated! It should be the sum of the compounds 1-6. At the same time, the Authors presented some data, but not connected with the Table content. It leads to incorrect conclusions on lines 336-348 and 371-373.
The lines 374-381 are not apparent to the reader. Therefore, it should be divided into two or more sentences.
The above-presented comments did not diminish the great work done during experiments but clarified the manuscript for the reader.
The presented manuscript should be improved and reviewed again.
Author Response
Please see the attachment。

Reviewer 2 Report
An extensive study was made by the authors to evaluate hepatotoxicity of dianthrones in HepG2 cells. Their findings suggest that dianthrones may be responsible for hepatotoxicity rather than anthrones as shown by other studies. In the manuscript, they describe optimization of the methods for extraction of dianthrones from Polygonum multiflorum Thunb. samples (were these root samples?) and separation of dianthrones. Their results of cell survival rate suggest that main hepatotoxic ingredients in the samples were cis- and trans- emodin-pyscion dianthrones.
However, while the results are interesting, some errors were found in the reported results. Thus, the interpretation of (some) results might have been done using incorrect results. Some of the results in the tables presenting the content of the analytes in the samples are incorrect (Tables 8-10). The calculated “Total” content is incorrect for several samples (e.g. Table 8: Sample PMR-01,PMR-56,..;Table 10: Sample PMR-12 b,PMRP-Q8h b, …). It is possible that also the reported contents for individual analytes are incorrect. The results and conclusions are therefore unreliable. All the tables and results should be checked and necessary corrections should be made.
I suggest that all large tables are moved to Supplementary material (Tables 1, 2, 8, 9, 10).
Figure 4 is not readable. Use higher resolution and larger fonts.
What is the purpose of using soya-bean milk (page 7, line 86)?
Why did you use smaller concentrations for trans- and cis- pyhsicon-physicon compounds (page 9, line 148)?
Clarify the purpose of using hydrochloric acid (page 10, line 200). What happens with your samples during the hydrolysis? What are free and bound dianthrone (page 18, line 312)?
Page 11, line 231- use “inter-day” instead of “intra-day” (as n=3 days).
Tables 4 and 5: There is no table caption to explain the meaning of the table values. Write the equations that you used to calculate the values in the table. What does No. 1 stands for? How are “Original”, “Spiked” and “Found” related? How do you calculate recovery?
Add captions to all tables and figures where missing.
Round 2
Reviewer 1 Report
The introduced changes and presenting some tables as supporting material make the manuscript clear and easy to read and understand. The reviewer's suggestions were submitted to the text, and questions were explained in the cover letter.
The manuscript can be accepted to be published in its present form.
Reviewer 2 Report
The authors addressed all comments in timely manner.